# Comparison of Surgical Outcomes of Adnexectomy by Vaginal Natural Orifice Transluminal Endoscopic Surgery (vNOTES) Versus Single-Port Access (SPA) Surgery

**DOI:** 10.3390/jpm12121996

**Published:** 2022-12-02

**Authors:** Jihee Jung, Joseph J. Noh, Jungeun Jeon, Chi-Son Chang, Tae-Joong Kim

**Affiliations:** 1Gynecologic Cancer Center, Department of Obstetrics and Gynecology, Samsung Medical Center, School of Medicine, Sungkyunkwan University, Seoul 06351, Republic of Korea; 2Department of Obstetrics and Gynecology, College of Medicine, Chung-Ang University, Seoul 06974, Republic of Korea

**Keywords:** natural orifice transluminal endoscopic surgery, single-port access surgery, gynecology, adnexectomy

## Abstract

The objective of the present pilot study is to compare operative outcomes between vaginal natural orifice transluminal endoscopic surgery (vNOTES) and single-port access (SPA) adnexectomy. Subjects were patients who underwent adnexectomy for benign adnexal disease, from November 2019 to May 2021. A total of 12 patients underwent vNOTES adnexectomy, and 55 patients received SPA laparoscopic adnexectomy. All surgeries were performed by one surgeon. In order to balance the baseline characteristics of the patients, 1:2 matching was performed. The vNOTES group demonstrated a low postoperative pain score within 12 h after surgery. They also required less use of analgesic medications postoperatively. Other surgical outcomes were comparable between the two groups. This study showed that vNOTES adnexectomy has comparable surgical outcomes to SPA.

## 1. Introduction

As a minimally invasive surgery, single-port access (SPA) and natural orifice transluminal endoscopic surgery (NOTES) have gained popularity in the field of gynecology in recent years [1,2]. SPA laparoscopic surgery is an endoscopic surgery performed through the umbilicus, an embryologic natural orifice. It has been reported to produce less postoperative pain and a higher cosmetic effect than that of conventional laparoscopy [3]. NOTES is an operation using natural openings such as the oral cavity, anus, urethra, or vagina [4]. Among these, surgery through the vagina began in gynecology and showed satisfactory results in many patients. In particular, the surgical approach through the vagina does not leave any wound on the abdomen, which is not only cosmetically excellent but also prevents the occurrence of potential adverse events, such as a postoperative wound infection or an abdominal wall hernia [5].

Vaginal natural orifice transluminal endoscopic surgery (vNOTES) leaves no scars from surgery, and it allows surgeons to access the ovaries and appendages, that are inaccessible in traditional vaginal approach surgery. It also allows visual evaluation of the entire abdominal cavity, including the upper parts including the liver and the diaphragm. Our group has performed SPA laparoscopy since 2008 and started vNOTES in 2018. Data on the safety and surgical outcomes of vNOTES are limited in the current literature [6,7]. The purpose of this pilot study is to compare surgical outcomes of vNOTES and SPA for adnexal surgery.

## 2. Materials and Methods

### 2.1. Patients

This study is a retrospective study, evaluating patients who underwent surgical procedures for benign ovarian disease from November 2019 to May 2021 at Samsung Medical Center, Seoul, South Korea. Patients with benign ovarian tumors or who needed prophylactic oophorectomy were included. During this period, 12 patients underwent vNOTES adnexectomy and 55 patients underwent SPA laparoscopic adnexectomy. In the case of endometriosis patients, those who were expected to have intraperitoneal adhesions or posterior cul-de-sac (PCDS) adhesions were excluded.

Blood tests and imaging tests were conducted prior to surgery. Imaging tests included ultrasound, computed tomography (CT), and magnetic resonance imaging (MRI). Patients with suspected endometriosis with severe intra-abdominal adhesions on preoperative images or who underwent combined hysterectomy were excluded from the study. The surgeon (TJ Kim) performed a bimanual pelvic examination to determine the feasibility of vaginal approach by assessing the anatomic conditions of the vagina such as the vaginal width. Those who had a history of vaginal delivery and who were not expected to have intra-abdominal adhesions were preferentially selected for vNOTES, whereas those with narrow vaginal openings or who were expected to have intra-abdominal adhesions were assigned for SPA laparoscopic surgery.

### 2.2. Operative Techniques

All surgical procedures were performed by an experienced single surgeon (TJ KIM). Patients underwent the same standard preparations before surgery. Prophylactic antibiotics were administered 30 min before the incision. Under general anesthesia, the patient was placed in the dorsal lithotomy position, and draping was done. Bladder drainage was done with a nelaton catheter. vNOTES surgical procedures were conducted as previously described [8]. For vNOTES adnexectomy, the cervix was grasped with two tenaculum forceps. Then, we opened the posterior vaginal wall using the Bovie electrocautery. The opening was usually 2.5–3.0 cm in length. After checking the posterior cul-de sac, we inserted a single-port platform (Glove Port^TM^, Neils, Inc., Bucheon, South Korea), which was the same as the single-port platform used for SPA adnexectomy. A pneumoperitoneum was established with CO_2_ at 10 or 11 mmHg, which was lower than that used in conventional laparoscopic surgery (Figure 1). The instruments used during the operations included a 5-mm rigid 30° camera, straight graspers (Ethicon Endosurgery, Inc., Cincinnati, OH, USA), and laparoscopic energy devices such as Enseal G2 Tissue Sealer^TM^ (Ethicon, Inc., Somerville, NJ, USA) and Caiman^TM^ (Aesculap AG, Inc., Tuttlingen, Germany). The procedures of SPA laparoscopic surgery were adopted from our previous study [9].

### 2.3. Data

The baseline clinical characteristics of the patients, including the results of the preoperative imaging studies, were collected. The primary outcome measure was operation time. The operation time was defined as the time from skin (or vaginal) incision to skin (or vaginal) suture. The secondary outcomes were the postoperative pain score and postoperative complication rates. The postoperative pain score was measured from 0 to 10 using the visual analog scale (VAS). The VAS score was measured and recorded immediately after surgery: 6 h after surgery, 12 h after surgery, and 24 h after surgery. Including intraoperative complications such as bowel, ureter, and bladder injury, we monitored whether immediate complications such as wound dehiscence and surgical site infection occurred within 6 weeks after surgery. Other measures included the estimated blood loss (EBL); hemoglobin changes between before and after surgery; the number of intravenous analgesics administered on postoperative day 1, in addition to the routine postoperative analgesic medications given to all patients; and the length of the postoperative hospital stay. Blood tests were performed on postoperative day 1, and the hemoglobin level was measured. All patients were administered nonsteroidal anti-inflammatory drugs (NSAIDs) and acetaminophen intravenously on the day of surgery. From postoperative day 1, they were administered oral NSAIDs. Frequency of analgesic medication uses were recorded.

### 2.4. Statistical Analyses

The data were expressed as mean ± standard deviation for continuous variables. Statistical significance was determined using *the Fisher’s exact test* for dichotomous variables and *the independent Student’s t-test* for continuous variables. We conducted a 1:2 propensity score matching to minimize the effect of baseline characteristics (age, parity, history of vaginal delivery, menopause status, abdominal surgery history, lesion size, and the side of surgical site) between the groups. Statistical significance was set at *p*-value < 0.05. The statistical calculations were performed with R version 4.2.0 (R Foundation for Statistical Computing, Vienna, Austria)

### 2.5. Ethics Approval

This study was performed in accordance with the ethical standards of the institution and with the 1964 Helsinki Declaration (and its later amendments). The Samsung Medical Center institutional review board approved the study.

## 3. Results

A total of 12 patients underwent vNOTES adnexectomy, and 55 patients underwent SPA laparoscopic adnexectomy during the study period. The baseline characteristics of all patients before and after propensity score matching are shown in Table 1. Before matching, there were statistically significant differences between the two groups in age (*p* = 0.001), vaginal delivery (*p* = 0.013), and menopause (*p* = 0.006). The groups were 1:2 matched to minimize the difference in baseline characteristics within *p* < 0.05 differences for all variables (age, parity, history of vaginal delivery, menopause status, body mass index (BMI), history of abdominal surgery, preoperative lesion size confirmed by imagings, and the side of surgical site).

The surgical information and postoperative outcomes are described in Table 2. There was no statistical difference in operation time between the SPA laparoscopy and vNOTES groups. Postoperative pain was reported to be lower in patients who underwent vNOTES adnexectomy compared with those who underwent SPA surgery. This pattern of postoperative pain was also observed even before statistical matching procedures were performed. In patients who underwent vNOTES, the pain score was significantly lower 12 h after surgery (*p* = 0.002). In regards to surgical complications, in one of the patients who underwent vNOTES, a small bowel thermal injury occurred from a laparoscopic energy device; therefore, an immediate reinforcement suture was performed. No additional complications were identified. Patients who underwent SPA adnexectomy did not have any case of conversion to other approaches. One patient in the vNOTES adnexectomy group was converted to SPA adnexectomy. In that patient, PCDS tissue thickening and bowel adhesion was noted, and a vaginal incision of 3 cm was not enough to approach the surgical site. EBL, hemoglobin changes, duration of hospital stay, and weight of the extracted specimen did not show any statistical difference between the two groups.

## 4. Discussion

The present study demonstrated that vNOTES adnexectomy is comparable to SPA laparoscopic adnexectomy for benign adnexal disease. As this study is a retrospective analysis, 1:2 matching was performed to minimize the difference in clinical baseline characteristics between the two groups according to the selection of subjects before surgery. There was no difference in operation time, but patients who underwent vNOTES had lower pain scores and fewer analgesic administrations.

The results of the present study are similar to those of previous studies comparing vNOTES and multiport laparoscopic surgery. According to previous studies, vNOTES surgery has been reported to be excellent at reducing postoperative pain with rapid recovery, lowering postoperative wound infection rates, and having excellent cosmetic effects [10]. The NOTABLE study was a single-center randomized, controlled trial and reported that vNOTES adnexectomy had a shorter operation time and lower postoperative pain [7]. Similarly, a Taiwan group reported that the transvaginal surgical approach resulted in significantly shorter operation time, less blood loss, and shorter hospital stay after surgery, resulting in better surgical outcomes [11].

In the present study, there was no significant difference in operation time between the SPA laparoscopy group and the vNOTES group. In a prospective study conducted at our institution comparing vNOTES hysterectomy and SPA hysterectomy, the operation time was shorter in the patients who underwent vNOTES [12]. When performing hysterectomy, the SPA laparoscopy group has to make two incision sites in the umbilicus and the vagina, but in vNOTES, incision through the umbilicus is not performed, thereby reducing the operation time. In contrast, vNOTES had no advantage in terms of operation time when performing adnexectomy. This study was performed as a pilot study, and the operation time was relatively long in the vNOTES group. A potential reason for this is that this study was performed in an early stage of our surgical skill development for vNOTES adnexectomy. Another potential reason for this observation is the fact that one patient in the vNOTES group converted to SPA laparoscopy. Initially, operation time was longer in the vNOTES group because it took longer to resolve bowel interference during the surgery. To solve this problem, surgical gauze was placed around the ovaries and fallopian tubes to secure a surgical field and prevent heat damage on the surrounding intestines (Figure 2).

Although there was no difference in pain level after 24 h, pain level within the first 12 h after surgery was significantly lower in the vNOTES group. The reason why there was less pain immediately after surgery is thought to be the fact that there is no abdominal incision in vNOTES. In addition to not having to incise the umbilicus, which has more nerve endings and sensory innervations than the vaginal fornix [13], relatively low CO_2_ gas exposure may have contributed to lowering postoperative pain.

This study was conducted on a limited number of patients, and the patients with severe adhesions due to endometriosis or who underwent hysterectomy were excluded. In this study, operation time for each surgical procedure (such as incision, ligation of the gonadal vessels, ligation of utero-ovarian vessels, and closure) was not analyzed separately, but the total operation time was analyzed. Therefore, it is suggested for future studies to analyze operation time for each procedure.

Ideal vNOTES patients are women who are amenable to vaginal access surgery through the vaginal cavity, who have a history of vaginal delivery, and who do not have intra-abdominal adhesions or any history of disease that might have induced adhesion. On the other hand, SPA laparoscopy is operable in all women. Therefore, vNOTES surgery should be applied to women who are sensitive to pain or those who want to avoid an abdominal wall surgical scar because of hypertrophic scars or keloids or who are judged to have easy suffocation access by examination. This examination can be done by a surgeon’s bimanual pelvic examination and preoperative imaging evaluations such as MRI and three-dimensional ultrasonography. If patients who can successfully undergo adnexectomy with a vaginal approach are carefully selected, there is less pain and high wound satisfaction during the recovery process after surgery.

Patient satisfaction, especially for benign gynecologic diseases in young women, is an important factor to consider when selecting surgical methods. In the present study, we did not assess patient satisfaction. However, based on the authors’ previous work [8], in which patient satisfaction after vNOTES hysterectomy was assessed by distributing a questionnaire, we think patient satisfaction in the vNOTES group of the present study is at least as high as that of those in the SPA group. This area should further be explored in future studies.

This pilot study showed that vNOTES is not inferior to SPA laparoscopy surgery when adnexectomy is performed. This study is a retrospective study that compares the initial data of vNOTES adnexectomy with SPA laparoscopic adnexectomy performed during the same period. There was no difference in the surgical results such as operation time and complications, but less pain was observed in vNOTES group. As a relatively safe and feasible method in benign adnexal diseases, it can be considered feasible. However, there are limitations in interpreting the results because of the small sample size of this study. The selection of the surgical approach was also based on the surgeon’s decision. Therefore, the limitation by the nature of a nonrandomized study was also present. It also carries limitations in that all surgical procedures were performed by one experienced surgeon. This may limit the generalizability of the study findings to others in different clinical settings. A large-scale prospective comparative study is currently underway to overcome this. Future studies are recommended to explore the learning curves of different minimally invasive approaches. It is also suggested that others explore the ergonomics and physical fatigues of surgeons performing different surgical techniques.

## Figures and Tables

**Figure 1 jpm-12-01996-f001:**
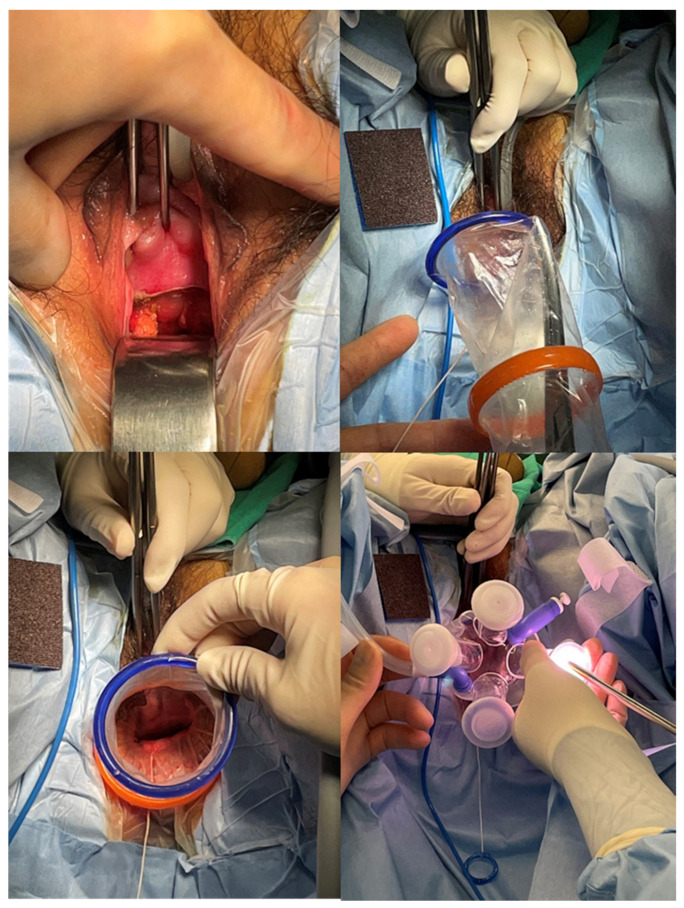
Procedures of vNOTES adnexectomy. The anterior and posterior lips of the cervix are grasped together with two tenaculum forceps. The incision is made at the posterior vaginal wall using the Bovie electrocautery. Using the curved Kelly forceps, the wound retractor is inserted into the posterior cul-de-sac and a single-port platform is installed.

**Figure 2 jpm-12-01996-f002:**
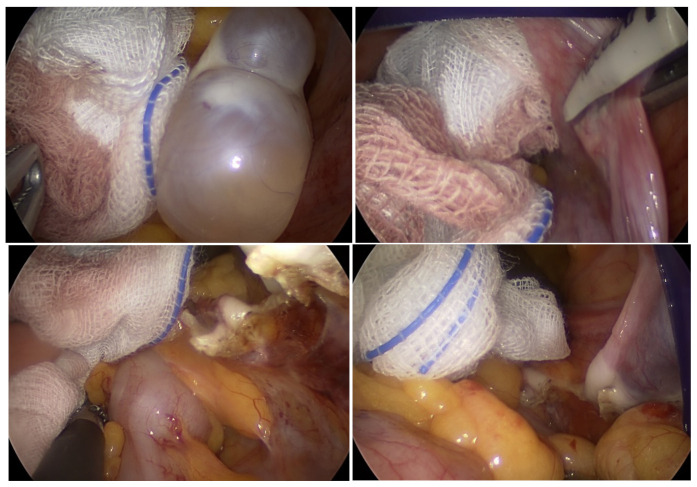
Because the intestine obscures the field of vision during surgery, gauze is inserted to secure the field of view.

**Table 1 jpm-12-01996-t001:** Baseline characteristics of the patients in vNOTES and SPA groups before and after matching.

	Before Matching			After Matching		
	vNOTES (*n* = 12)	SPA (*n* = 55)	*p*-Value	vNOTES (*n* = 12)	SPA (*n* = 24)	*p*-Value
Age (years)	62.5 [60.0; 67.0]	53.0 [46.5; 60.5]	0.001	62.7 ± 5.3	58.8 ± 5.6	0.055
Parity			0.282			0.432
0	0 (0.0%)	6 (10.9%)		0 (0.0%)	1 (4.2%)	
1	1 (8.3%)	10 (18.2%)		1 (8.3%)	4 (16.7%)	
2	6 (50.0%)	30 (54.5%)		6 (50.0%)	15 (62.5%)	
3	5 (41.7%)	7 (12.7%)		5 (41.7%)	4 (16.7%)	
4	0 (0.0%)	1 (1.8%)		0 (0.0%)	0 (0.0%)	
5	0 (0.0%)	1 (1.8%)		0 (0.0%)	0 (0.0%)	
History of vaginal delivery			0.013			1.000
No	0 (0.0%)	20 (36.4%)		0 (0.0%)	1 (4.2%)	
Yes	12 (100.0%)	35 (56.4%)		12 (100.0%)	23 (95.8%)	
Menopause			0.006			1.000
No	0 (0.0%)	22 (36.4%)		0 (0.0%)	1 (4.2%)	
Yes	12 (100.0%)	33 (63.6%)		12 (100.0%)	23 (95.8%)	
BMI ^a^ (kg/m^2^)	23.5 [22.3;25.2]	23.3 [21.8;25.0]	0.954	23.3 ± 2.4	23.6 ± 2.9	0.809
History of previous abdominal surgery			0.625			0.612
0	7 (58.3%)	29 (52.7%)		7 (58.3%)	16 (66.7%)	
1	5 (41.7%)	16 (29.1%)		5 (41.7%)	6 (25.0%)	
2	0 (0.0%)	8 (14.5%)		0 (0.0%)	1 (4.2%)	
3	0 (0.0%)	1 (1.8%)		0 (0.0%)	1 (4.2%)	
4	0 (0.0%)	1 (1.8%)		0 (0.0%)	0 (0.0%)	
Pre-op size (cm)	4.8 ± 2.1	5.6 ± 2.7	0.325	4.4 ± 2.1	5.9 ± 2.7	0.111
Operation side			0.314			0.253
USO ^b^	2 (16.7%)	19 (34.5%)		2 (16.7%)	1 (4.2%)	
BSO ^b^	10 (83.3%)	36 (65.5%)		10 (83.3%)	23 (95.8%)	

^a^ BMI: body mass index. ^b^ USO, unilateral salpingo-oophorectomy; BSO, bilateral salpingo-oophorectomy.

**Table 2 jpm-12-01996-t002:** Surgical and postoperative outcomes of the vNOTES and SPA adnexectomy before and after matching.

	Before Matching			After Matching		
	vNOTES (*n* = 12)	SPA (*n* = 55)	*p*-Value	vNOTES (*n* = 12)	SPA (*n* = 24)	*p*-Value
Operation time (min)	52.5 [38.5; 64.0]	48.0 [41.0; 60.0]	0.595	52.5 [38.5; 64.0]	48.5 [41.0; 58.0]	0.523
Estimated blood loss (mL)	50.0 [40.0; 50.0]	30.0 [30.0; 50.0]	0.186	50.0 [40.0; 50.0]	30.0 [20.0; 50.0]	0.148
Hemoglobin changes (g/dL) ^a^	1.1 ± 0.6	1.2 ± 0.7	0.681	1.1 ± 0.6	1.1 ± 0.8	0.895
Postoperative hospital stay (day)	1 (100.0%)	1 (100.0%)		1 (100.0%)	1 (100.0%)	
Conversion of surgical methods	1	0		1	0	
Postoperative pain score						
6 h	2.5 ± 1.0	3.8 ± 1.6	0.006	2.5 ± 1.0	3.6 ± 1.6	0.037
12 h	1.5 [0.0; 2.0]	3.0 [2.0; 3.0]	0.008	1.5 [0.0; 2.0]	3.0 [3.0; 5.0]	0.002
24 h	1.0 [0.0; 2.5]	2.0 [2.0; 3.0]	0.077	1.0 [0.0; 2.5]	3.0 [2.0; 3.0]	0.058
Analgesic requested on POD ^b^ 1			0.552			0.536
None	12 (100.0%)	45 (81.8%)		12 (100.0%)	21 (87.5%)	
IV acetaminophen	0 (0.0%)	10 (18.2%)		0 (0.0%)	3 (12.5%)	
Immediate postoperative complications	1	0		1	0	
Delayed postoperative complications	0	0		0	0	

^a^ Hemoglobin on postoperative day 1 minus preoperative hemoglobin. ^b^ POD, postoperative day.

## Data Availability

The data presented in this study are available on request from the corresponding author.

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
