# Peer review of "Comparison of Surgical Outcomes of Adnexectomy by Vaginal Natural Orifice Transluminal Endoscopic Surgery (vNOTES) Versus Single-Port Access (SPA) Surgery"

_jpm, 2022, doi:10.3390/jpm12121996_

Round 1

Reviewer 1 Report

Although the study is retrospective but It still has significance to the literature.

The manuscript can be improved on the number of fronts:

1.In terms of the surgical time being statistically not significant, one can compare the cases performed in learning curve for single port surgery vs vNOTES (maybe the first ten, twelve cases)

2.MRI or 3D ultrasound can provide information about obliterated POD why was this not have been considered to decide upon the technique of the surgery?

3.There is no comment on comment satisfaction in both groups

4.There is no comment on the impact of the surgery on the surgeon considering the ergonomics of the two different techniques.

5.There is no mention about visual pain score. How did the patients qualify for analgesia in immediate post-op and one week later?

Author Response

  1. MRI or 3D ultrasound can provide information about obliterated POD why was this not have been considered to decide upon the technique of the surgery?
    Thank you for your comment. As you said, magnetic resonance imaging (MRI) is a sensitive imaging modality to evaluate the extent of pelvic adhesion. However, not all patients with benign gynecologic disease can undergo MRI evaluation due to its high cost and time. In the present institution, ultrasonography is preferentially performed because of its easy and quick accessibility and low cost (compared to MRI). For those patients who are suspected to have deep infiltrating endometriosis or those who require further imaging evaluation in addition to ultrasonography are selectively recommended to undergo MRI evaluation. In the present study, if any signs of pelvic adhesion were seen on MRI evaluation (or ultrasonography), we excluded those patients for vNOTES. The attending surgeon also performed bimanual pelvic examinations to determine the extent of possible adhesion in the pelvic cavity.
    3D ultrasonography is also a sensitive imaging tool to evaluate pelvic adhesion or deep infiltrating endometriosis. The present institution is equipped with 3D ultrasonography. However, we only perform 3D ultrasonography in certain cases, such as during the assessment of uterine anomaly, due to its high cost and medical insurance coverage regulation. 3D ultrasonography is still at its early stage in our day-to-day practice.
    Even though we did not utilize 3D ultrasonography in the present study, we agree with the reviewer’s opinion in that it can be a useful preoperative imaging tool to evaluate the extent of pelvic adhesion in patients with benign gynecologic diseases. Therefore, we added a sentence in the Discussion section to highlight the potential utilization of 3D ultrasonography (Lines 214 – 216).
  2. There is no comment on comment satisfaction in both groups
    Patient satisfaction especially for benign gynecologic diseases in young women is an important factor to consider when selecting surgical methods. Unfortunately, we did not assess patient satisfaction in the present study. We think this is an important area that needs further evaluation in future studies.
    Based on our previous work (Reference No. 8 in our manuscript, Frontiers in Medicine, 2021), in which patient satisfaction was assessed by a questionnaire, we think the patient satisfaction in the vNOTES group of the present study is at least as high as those in the SPA group. However, as the reviewer commented, the direct comparison of patient satisfaction between vNTOES and SPA should further be explored and we commented this in the Discussion session (Lines 219 – 225).
  3. There is no comment on the impact of the surgery on the surgeon considering the ergonomics of the two different techniques.
    Thank you for your valuable comment. Surgeon’s ergonomics is definitely a crucial factor when comparing two different surgical approaches. Because this was a pilot study with a small number of patients and all surgical procedures were performed by a single surgeon, we thought that it would be too early to mention surgeon’s ergonomics in the present study. However, because we strongly agree with the reviewer’s opinion, we will assess this aspect in our future studies. We also mentioned this in Lines 239 – 240.
  4. There is no mention about visual pain score. How did the patients qualify for analgesia in immediate post-op and one week later?
    As it is widely recognized, pain is a very subjective matter and it is very difficult to objectively measure pain. Comparing pain between two individuals is also not feasible. Therefore, in the present study, as asked the patients to request analgesic medication if they want to receive rescue analgesia in addition to the routine pain medication (Lines 104 – 107).
    In order to clarify this further, we added an additional explanation in Lines 102 – 103.
  5. In terms of the surgical time being statistically not significant, one can compare the cases performed in learning curve for single port surgery vs vNOTES (maybe the first ten, twelve cases).
    Thank you for your valuable comment. We agree that comparison between the early phase of SPA surgeries and that of vNOTES can provide further objective measures to assess the learning curve of this new surgical technique. We, in deed, are in preparation for analyzing the learning curves of SPA surgeries and of vNOTES surgeries. We started SPA surgeries in 2008 and vNOTES in 2018. For vNOTES, we performed vNOTES hysterectomy first, then started to perform vNOTES adnexectomy after we reached our proficiency levels with opening of the vagina. We previously reported our early experience (learning curve) of SPA surgeries in 2011 (Eur J Obstet Gynecol Reprod Biol, 2011, DOI: 10.1016/j.ejogrb.2011.04.017). We also reported our early experience of vNOTES in 2021 (Front Med, 2021, DOI: 10.3389/fmed.2020.583147). In our future studies that we are currently working on, we will report the results of the comparison between our early experiences with SPA and vNOTES surgeries in terms of the number of surgical procedures required to reach the proficiency (plateau) levels, surgical time, and incidences of complications.

Reviewer 2 Report

The manuscript is clear, understandable, the methods are well described, the results follow the methods and is ethically acceptable. As I am not a native English speaker I am not able to comment on grammar, however from my perspective the language is completely understandable.

In general the manuscript shows additional information about the safety of vNOTES in comparison to SPA on a small number of patients when the method was introduced in a single centre by a single surgeon. The results confirm the safety of the method in this circumstances. As is cited in the manuscript the safety was however already recognised in a meta-analysis on higher number of patients. The strongest point of presented manuscript is therefore a confirmation that it is safe to start with the procedure if the surgeon is already experienced in SPA and has an access to the equipment. This could be a major conclusion of the manuscript.

It would improve the manuscript if authors would present more detailed information about what kind of experience are necessary for a surgeon to be qualified to start with the procedure in their own institution (how many procedures are performed per year, what kind of procedures, etc). And to be discussed why only one surgeon in this centre was included and whether vNOTES is not safe to be performed by an average skilled surgeon and why. The new methods might be more interesting to other surgeons if they are feasible to all surgeons performing such frequent procedures as are removals of ovaries. The comment about the learning curve would be welcomed. 

It is logical that the two groups of patients differ in number of vaginal deliveries, menopause and age, as these are directly or indirectly associated with selection criteria for vNOTES. This fact could be commented with an emphasise on a conclusion that approach is associated with patient selection that reduces the generalisation of the method.

What is missing is also patients' reported outcomes to evaluate the subjective feed back from the patients. To learn whether the observed results are clinically relevant from patients' perspective and justify the method. If not available now, I would strongly suggest to include this for further research. 

Author Response

  1. The manuscript is clear, understandable, the methods are well described, the results follow the methods and is ethically acceptable. As I am not a native English speaker I am not able to comment on grammar, however from my perspective the language is completely understandable.
    Thank you for your valuable time reviewing our manuscript.
  2. In general the manuscript shows additional information about the safety of vNOTES in comparison to SPA on a small number of patients when the method was introduced in a single centre by a single surgeon. The results confirm the safety of the method in this circumstances. As is cited in the manuscript the safety was however already recognised in a meta-analysis on higher number of patients. The strongest point of presented manuscript is therefore a confirmation that it is safe to start with the procedure if the surgeon is already experienced in SPA and has an access to the equipment. This could be a major conclusion of the manuscript.
    Yes, we appreciate your concise summary and agree with your interpretation.
  3. It would improve the manuscript if authors would present more detailed information about what kind of experience are necessary for a surgeon to be qualified to start with the procedure in their own institution (how many procedures are performed per year, what kind of procedures, etc). And to be discussed why only one surgeon in this centre was included and whether vNOTES is not safe to be performed by an average skilled surgeon and why. The new methods might be more interesting to other surgeons if they are feasible to all surgeons performing such frequent procedures as are removals of ovaries. The comment about the learning curve would be welcomed.
    Thank you for your comment. As you mentioned, one major drawback of the present study is the fact that all surgical procedures were performed by a single surgeon. Comparison of surgical outcomes between SPA and vNOTES may be more accurately portrayed if a single surgeon performs all SPA and vNOTES surgeries such as the present study, but this certainly limits the generalization of the findings to other typical surgeons in other clinical settings.
    The attending surgeon in the present study has an ample experience in conventional laparoscopy, single-port access (SPA) laparoscopy, robotic surgeries, and vaginal approach surgeries. He’s been practicing gynecologic oncology for more than 15 years and performs about 40 minimally-invasive surgeries per month. His substantial amount of experience in surgical procedures may limit the generalizability of the study findings to others. We mentioned this limitation in Lines 234 – 236.
    Performing an accurate assessment of learning curve when a new surgical technique is developed is crucial. We, in deed, are in preparation for analyzing the learning curves of SPA surgeries and of vNOTES surgeries. We started SPA surgeries in 2008 and vNOTES in 2018. For vNOTES, we performed vNOTES hysterectomy first, then started to perform vNOTES adnexectomy after we reached our proficiency levels with opening of the vagina. We previously reported our early experience (learning curve) of SPA surgeries in 2011 (Eur J Obstet Gynecol Reprod Biol, 2011, DOI: 10.1016/j.ejogrb.2011.04.017). We also reported our early experience of vNOTES in 2021 (Front Med, 2021, DOI: 10.3389/fmed.2020.583147). In our future studies that we are currently working on, we will report the results of the comparison between our early experiences with SPA and vNOTES surgeries in terms of the number of surgical procedures required to reach the proficiency (plateau) levels, surgical time, and incidences of complications.
  4. It is logical that the two groups of patients differ in number of vaginal deliveries, menopause and age, as these are directly or indirectly associated with selection criteria for vNOTES. This fact could be commented with an emphasise on a conclusion that approach is associated with patient selection that reduces the generalisation of the method.
    Thank you. We agree with your comment. The fact that the type of surgery was decided by the surgeon certainly limits the generalizability. The study nature of a non-randomized design carries limitation. We emphasized this limited generalizability of the study findings in Lines 232 – 234.
  5. What is missing is also patients' reported outcomes to evaluate the subjective feed back from the patients. To learn whether the observed results are clinically relevant from patients' perspective and justify the method. If not available now, I would strongly suggest to include this for further research.
    Patient reported outcome (PRO) is a valuable tool to collect important information that may otherwise be missed by traditional data collecting methods. We think PRO may especially be useful for collecting patient overall satisfaction and pain. Patient satisfaction especially for benign gynecologic diseases in young women is an important factor to consider when selecting surgical methods. Unfortunately, we did not assess patient satisfaction or pain by PRO in the present study. We think this is an important area that needs further evaluation in future studies as you commented.
    Based on our previous work (Reference No. 8 in our manuscript, Frontiers in Medicine, 2021), in which patient satisfaction was assessed by a questionnaire (not PRO), we think the patient satisfaction in the vNOTES group of the present study is at least as high as those in the SPA group. We commented this in the Discussion session (Lines 219 – 225).